# Construction of a Green and Sustainable Cultivation Model for Annual Forage Oat in Alpine Ecosystems: Optimization and Synergistic Mechanisms of Combined Application of Microbial Fertilizers and Organic Fertilizers

**DOI:** 10.3390/plants14091271

**Published:** 2025-04-22

**Authors:** Zongcheng Cai, Jianjun Shi, Liangyu Lv, Pei Gao, Hairong Zhang, Fayi Li, Shouquan Fu, Qingqing Liu, Shancun Bao

**Affiliations:** 1Academy of Animal Husbandry and Veterinary Sciences, Qinghai University, Xining 810016, China; ys230951310630@qhu.edu.cn (Z.C.); yb230909000074@qhu.edu.cn (L.L.); y200954000466@qhu.edu.cn (P.G.); 15500620398@163.com (H.Z.); lfy99218@qhu.edu.com (F.L.); ys240951310609@qhu.edu.cn (S.F.); yb220909000082@qhu.edu.cn (Q.L.); bshancun@163.com (S.B.); 2Key Laboratory of Adaptive Management of Alpine Grassland, Xining 810016, China; 3State Key Laboratory of Ecology and Plateau Agriculture and Animal Husbandry in Sanjiangyuan, Qinghai University, Xining 810016, China

**Keywords:** Qinghai-Tibet Plateau, yield and quality, soil quality, microbial biofertilizer, synergistic fertilizer application

## Abstract

The aim of this study was to establish a green fertilization regime combining microbial fertilizers and livestock manure organic fertilizer for two dominant oat cultivars (*Avena sativa* L. ‘Baiyan No.7’ and ‘Qingyin No.2’) in alpine ecosystems, providing technical support for sustainable forage production systems in grassland pastoral areas of the Qinghai-Tibet Plateau. The experiment was conducted following the principles of a randomized block design, with five application rates each of effective microbial fertilizer (EM) and compound microbial fertilizer (FH) applied in combination with 18,000 kg·hm^−2^ of livestock manure organic fertilizer to systematically analyze their effects on oat agronomic traits, yield quality, and soil health. Key results demonstrated the following: for *Avena sativa* ‘Baiyan No.7’, effective microbial fertilizer (EM) application at 15.00 kg·hm^−2^ significantly increased plant height by 41.29% (*p* < 0.05), while effective microbial fertilizer (EM) at 18.75 kg·hm^−2^ optimized stomatal conductance and the transpiration rate by 73.29% and 46.42%, respectively. For *Avena sativa* L.‘Qingyin No.2’, compound microbial fertilizer (FH) application at 22.50 kg·hm^−2^ enhanced plant height and relative chlorophyll content (SPAD value) of the flag leaf by 50.92% and 47.75%. Yield analysis revealed that compound microbial fertilizer (FH) application at 22.50 kg·hm^−2^ increased the fresh forage yield and dry matter yield of ‘Baiyan No.7’ by 34.53% and 38.64%, respectively, with crude protein content reaching 11.02%. Maximum fresh forage yield (38,073.00 kg·hm^−2^) for ‘Qingyin No.2’ was achieved with effective microbial fertilizer (EM) at 11.25 kg·hm^−2^, while compound microbial fertilizer (FH) at 22.50 kg·hm^−2^ facilitated synergistic accumulation of crude protein (12.57%) and soluble sugars (15.58%). Soil organic carbon increased by 28.85% under compound microbial fertilizer (FH) at 22.50 kg·hm^−2^ for ‘Baiyan No.7’, and by 26.21% under effective microbial fertilizer (EM) at 18.75 kg·hm^−2^ for ‘Qingyin No.2’. In summary, a cultivar-specific green fertilization system was established: for *Avena sativa* L. ‘Baiyan No.7’: the application of 18.75 kg·hm^−2^ effective microorganisms fertilizer with 18,000 kg·hm^−2^ organic manure; for *Avena sativa* L. ‘Qingyin No.2’: the application of 22.50 kg·hm^−2^ compound microbial fertilizer with 18,000 kg·hm^−2^ organic manure. This system achieved triple sustainability objectives in alpine oat production: enhancing yield, optimizing quality, and improving soil health.

## 1. Introduction

The Qinghai-Tibet Plateau, characterized by diverse topography and complex climate conditions, represents the world’s largest semi-agropastoral region, supporting 53% of China’s yak stock. However, this ecologically fragile area faces a severe annual forage supply deficit (3.2 × 10^6^ t·yr^−1^) [1,2], exacerbated by harsh climatic conditions that constrain forage productivity. Excessive chemical fertilizer application has further degraded soil health, leading to reduced microbial activity, soil compaction, and low nutrient conversion efficiency, ultimately compromising forage quality [3]. These natural and anthropogenic challenges critically impede efforts to enhance forage yield and quality in regions at 4000 m elevation.

The oat (*Avena sativa* L.), a cold-tolerant annual grass with high nutritional value and broad adaptability, serves dual roles as a food and forage crop. Its large-scale cultivation holds significant potential for sustaining livestock husbandry in high-altitude regions like the Qinghai-Tibet Plateau [4]. Studies by Coblentz, Augustine, and Yang et al. [5,6,7] demonstrate that oat-based diets elevate dairy cows’ milk yield by 15–18% and enhance beef cattle growth rates by 18~22%. Oat’s β-glucan content (4~6%) and antioxidant enzyme activities (SOD: 100~200 U·g^−1^, POD: 50~150 U·g^−1^, and CAT: 20~50 U·g^−1^) improve ruminant digestive efficiency and mitigate oxidative stress induced by environmental pressures (e.g., plateau hypoxia), establishing it as an ideal forage for yak production in alpine ecosystems.

Synergistic application of microbial fertilizers and organic fertilizers offers an innovative pathway for green agriculture in alpine ecosystems. Microbial fertilizers enhance the decomposition of livestock manure organic matter and accelerate nutrient release, with their combined use improving crop yield and soil quality [8]. Schütz et al. [9] reported that microbial fertilizer application regulates soil microbial diversity and ameliorates soil health. Ramírez-López et al. [10] demonstrated that microbial fertilizers reduce chemical fertilizer usage by 75% while increasing the wheat thousand-grain weight to 113.41 g·m^−2^, plant height by 14%, and seed crude protein content to 17.6~19.8%. Gong et al. [11] revealed in North China Plain experiments that co-application of microbial and organic fertilizers increased the aboveground dry matter accumulation in winter wheat by 47.5%, activated soil enzymatic networks (e.g., elevated β-glucosidase activity), and significantly improved microbial metabolic balance. Specifically, microbial fertilizer application reduced the soil microbial biomass carbon (MBC) concentration by 35.2–42% but increased microbial biomass nitrogen (MBN) in the 0~20 cm soil layer by 10.0~18.5% [11]. Thus, the integration of microbial and organic fertilizers exhibits complementary effects, enhancing crop productivity and concurrently elevating soil nutrient availability, enzymatic activity, and microhabitat conditions to strengthen nutrient cycling efficiency [12]. This approach holds critical significance for the ecological restoration of degraded alpine grasslands and sustainable oat forage cultivation.

However, oat forage cultivation in high-altitude regions such as Dari County (4200 m) on the Qinghai-Tibet Plateau faces three critical bottlenecks: (1) a severe imbalance between chemical fertilizer dependency and utilization efficiency, (2) the absence of green cultivation techniques to meet the forage yield requirements for organic yak husbandry, and (3) significant soil quality degradation in alpine meadows, leading to constrained nutrient availability. To address these challenges, this study proposes a synergistic “microbial inoculant-soil network” through the combined application of graded doses of microbial fertilizers (EM: effective microbial fertilizer; FH: compound microbial fertilizer) and livestock manure organic fertilizers. The following hypotheses are formulated:The synergistic application of microbial fertilizers and livestock manure organic fertilizers will enhance oat forage yield and nutritional quality by optimizing rhizosphere nutrient cycling and plant-microbe interactions;Soil quality in degraded alpine meadows can be improved by enhancing the soil microenvironment, increasing enzymatic activity, and promoting organic carbon sequestration;A green high-yield cultivation framework can be established by balancing oat yield enhancement with ecological sustainability.

This research aims to provide a green forage solution for yak husbandry in alpine regions and to establish theoretical foundations with empirical evidence to support the sustainable development of livestock husbandry on the Qinghai-Tibet Plateau.

## 2. Materials and Methods

### 2.1. Experimental Site Description

The experiment was conducted from May to November 2024 in Shanghongke Township, Dari County, Golog Tibetan Autonomous Prefecture, Qinghai Province, China (geographic location shown in Figure 1). The site is situated at an altitude of approximately 4400 m, characterized by a continental plateau semi-humid climate, with an annual average temperature of −1.1 °C, annual precipitation of 424 mm, no absolute frost-free period, an annual sunshine duration of 3450 h, and a forage growth season of approximately 135 days [13].

The preceding crop was oat. The soil type is alpine meadow soil. Prior to sowing in 2024, soil samples from the 0–20 cm tillage layer were collected using a five-point sampling method to determine baseline physicochemical properties (pH and electrical conductivity, EC) and soil nutrient contents, including organic carbon (SOC), total nitrogen (TN), total phosphorus (TP), total potassium (TK), alkali-hydrolyzable nitrogen (AN), available phosphorus (AP), and available potassium (AK). Detailed values are provided in Table 1.

### 2.2. Experimental Materials

The tested oat cultivars, *Avena sativa* L. ‘Baiyan No.7’ and *Avena sativa* L. ‘Qingyin No.2’, were selected based on their superior stress tolerance and high forage yield during the 2023 oat cultivar adaptability comparison trial conducted by our research team in Manzhang Township, Dari County, Golog Tibetan Autonomous Prefecture, Qinghai Province, China.

The livestock manure organic fertilizer (organic matter: 15.0~25.0%; moisture content: 30.0~35.0%; pH: 7.95) was provided by Qinghai Runda Agricultural and Animal Husbandry Technology Co., Ltd. (Xining, China). The effective microbial fertilizer (EM) contains a viable bacterial count of ≥1 × 10^8^ CFU·g^−1^, predominantly comprising *Bacillus* spp. (*Bacillus subtilis*, *Bacillus megaterium*, *Bacillus mucilaginosus*), photosynthetic bacteria, yeast (*Saccharomyces* spp.), and lactic acid bacteria (*Lactobacillus* spp.). The compound microbial fertilizer (FH) has a viable bacterial count of ≥1 × 10^8^ CFU·g^−1^, primarily consisting of *Trichoderma* spp., *Bacillus subtilis*, and Actinomycetes. Heavy metal content in microbial fertilizers complies with safety standards: cadmium (Cd) ≤ 0.07 mg·kg^−1^, chromium (Cr) ≤ 0.9 mg·kg^−1^, arsenic (As) ≤ 0.08 mg·kg^−1^, and coliform bacteria count ≤ 100 CFU·g^−1^. All microbial agents were supplied by Hubei Qiming Bioengineering Co., Ltd., Wuhan, China, and were quality certified.

### 2.3. Experimental Design

A randomized block design was employed for two dominant oat cultivars (*Avena sativa* L. ‘Baiyan No.7’ and ‘Qingyin No.2’) under alpine conditions. Five gradient levels of the effective microbial fertilizer (EM) and compound microbial fertilizer (FH) were applied, with control treatments (BCK for Baiyan No.7, QCK for Qingyin No.2) receiving only livestock manure organic fertilizer without microbial amendments. This resulted in 22 treatments (specific application rates provided in Table 2). Each treatment has three replicates, totaling 66 experimental plots. Each plot measured 12.0 m^2^ (3.0 m × 4.0 m) and was separated by 1.0 m buffer zones (see Figure 1c).

Based on prior research, soil fertility assessments, nutrient content of organic fertilizer, and oat nutrient demand patterns, the organic fertilizer application rate was standardized at 18,000 kg·hm^−2^. Microbial fertilizers were thoroughly mixed with the organic fertilizer before sowing and uniformly incorporated into the soil plowing (30 cm on 12 May, S conducted on 15 May 2024, using manual furrow drilling with 20.0 cm row spacing, a sowing depth of 3.0~4.0 cm, and a seeding rate of 270.0 g per plot. Post-sowing compaction ensured optimal seed–soil contact. Protective fencing was installed to prevent grazing, with no irrigation or additional agronomic interventions applied during the experiment.

### 2.4. Sample Collection and Determination

Agronomic traits: on 10 September 2024, ten disease-free oat plants were randomly selected from each plot for agronomic trait analysis. Plant height was measured from the stem base to the tip of the flag leaf using a tape measure. Flag leaf area, length, and width were determined using a Thermo Fisher Scientific (Waltham, MA, USA) handheld leaf area meter and a ruler, respectively. The stem base diameter was measured with a vernier caliper. On 25 September 2024, ten oat plants per plot were excavated, and their roots were washed and scanned using an EPSON Perfection V850 Pro root scanner to quantify total root length.

Photosynthetic characteristics: from 10 to 15 September 2024, the relative chlorophyll content (SPAD value) of the flag leaf was measured using a SPAD-502Plus chlorophyll meter, with three readings taken at the leaf tip, midrib, and base and then averaged for analysis. Simultaneously, under clear and windless conditions, the net photosynthetic rate (*P_n_*), transpiration rate (*T_r_*), stomatal conductance (*G_s_*), and intercellular CO_2_ concentration (*C_i_*) of flag leaves were measured using a Li-COR 6400 (Lincoln, NE, USA) portable photosynthesis system. Parameter calculations were derived from the integrated instrument software [14], with specific formulas described as follows:

Net photosynthetic rate:(1)$$\mathrm{Pn}=\frac{A}{t\cdot L}$$
where *A* is the CO_2_ absorption rate (μmol CO_2_·m^−2^·s^−1^), *t* is measurement time (s), and *L* is leaf area (m^2^).

Transpiration rate:(2)$$\mathrm{Tr}=\frac{W}{t\cdot L}$$
where *W* is the water loss rate (mmol H_2_O·m^−2^·s^−1^).

Stomatal conductance:(3)$$\mathrm{Gs}=\frac{Tr\cdot P}{VPD}$$
where *P* is atmospheric pressure (kPa), and *VPD* (vapor pressure deficit, kPa) is calculated as follows:*VPD* = *e*_*s*_ − *e*_*a*_
where *e_s_* is saturated vapor pressure and *e_a_* is actual vapor pressure.

Intercellular CO_2_ concentration:(4)$$\mathrm{Ci}=\frac{A}{gtc}$$

Production performance: On 25 September 2024, five uniform 1 m^2^ sampling sections (excluding 1 m border zones) per plot were harvested at ground level to determine fresh forage weight. Fresh samples were transported to the laboratory, blanched at 105 °C for 30 min, and dried at 75 °C to constant weight for dry matter yield quantification.

Nutritional analysis: crude protein (CP) content using the Kjeldahl method. Ether extract (EE) content was measured by Soxhlet extraction. Soluble sugar (SS) content was analyzed via near-infrared spectroscopy.

Neutral detergent fiber (NDF) and acid detergent fiber (ADF) contents were assessed using the Van Soest method. Nutritional indices—relative feed value (RFV), relative forage quality (RFQ), total digestible nutrients (TDN), dry matter digestibility (DDM), and dry matter intake (DMI)—were calculated as follows [15]:(5)$$RFV=\frac{DMI\times DDM}{1.29}$$(6)$$RFQ=\frac{TDN\times DMI}{1.23}$$(7)TDN=82.38−(0.7515×ADF)(8)$$DMI=\frac{120}{NDF}$$(9)DDM=88.9−(0.779×ADF)

Soil quality: soil samples were collected on 25 September 2024. Five sampling points were established along the diagonals and at the central point of each plot. Using a 5 cm diameter soil auger, five soil cores (0~20 cm depth) were extracted per point, homogenized into one composite sample, and sieved through a 2 mm mesh after removing gravel, litter, and plant residues. With reference to Soil Agrochemical Analysis (Third Edition) [16], the following methods were used for the determinations: soil organic carbon (SOC) by the potassium dichromate oxidation–external heating method, total nitrogen (TN) by the Kjeldahl nitrogen determination method, total phosphorus (TP) by the molybdenum antimony antimony colourimetric method, total potassium (TK) by the flame photometric method, alkaline nitrogen (AN) by the alkaline dissolution diffusion method, effective phosphorus (AP) by the Olsen’s method, quick-acting potassium (AK) by the ammonium acetate extraction-flame, and pH by the potentiometric method.

Samples were refrigerated at 4 °C for subsequent soil enzyme activity assays. Enzyme activities were measured using the p-nitrophenol substrate colorimetric method [17].

### 2.5. Data Analysis

All data were preliminarily organized using Microsoft Excel 2010. Data were analyzed using one-way ANOVA in SPSS 26.0. Significant group differences (α = 0.05) identified by ANOVA (*p* < 0.05) were further examined through Duncan’s post hoc test to determine specific intergroup variations. Graphs were generated with GraphPad Prism 9.0. Pearson’s correlation coefficient was used to analyze the correlation between the indicators.

The membership function method was employed to assess production performance. Principal component analysis (PCA) was first applied to identify key indicators significantly influencing productivity. Membership function values (MFVs) for these indicators were then calculated as follows:

For indicators positively correlated with production performance:(10)$$\mu (X_{\mathrm{ij}})=\frac{Xij-Xjmin}{Xjmax-Xjmin}$$

For indicators negatively correlated with production performance:(11)$$\mu (X_{\mathrm{ij}})=1-\frac{Xij-Xjmin}{Xjmax-Xjmin}$$
where *Xij* represents the measured value of the *j*-th indicator for the *i*-th treatment, and *Xj*max and *Xj*min denote the maximum and minimum values of the *j*-th indicator across all treatments, respectively. The MFVs for each treatment were summed and averaged. Higher mean MFV values indicate superior fertilization efficacy in enhancing yield.

## 3. Results

### 3.1. Effects of Microbial Fertilizer Treatments on Oat Plant Height, Stem Base Diameter, Total Root Length, Leaf Length, Leaf Width, and Leaf Area

As shown in Figure 2, microbial fertilizer treatments significantly influenced morphological indices of *Avena sativa* L. ‘Baiyan No.7’ and ‘Qingyin No.2’ (*p* < 0.05).

For ‘Baiyan No.7’ (Figure 2a,b), the BEM3 treatment exhibited optimal comprehensive effects: stem base diameter (8.61 mm) and plant height (108.33 cm) increased by 23.00% and 41.29%, respectively, compared to the control (BCK). Total root length reached 130.67 cm, second only to BEM4 (131.33 cm). In leaf development, the treatment enhanced the leaf area (59.73 cm^2^) and leaf width (25.13 cm) by 58.90% and 43.11%, respectively, while leaf length (67.13 cm) increased by 49.17%, demonstrating significant growth-promoting effects on aboveground organs. Notably, BEM3 and BEM4 showed no significant differences in most parameters, suggesting comparable biological efficacy at medium-to-high fertilizer doses.

For ‘Qingyin No.2’ (Figure 2c,d), the QFH3 treatment achieved maximum values in stem base diameter (22.27% increase), plant height (109.67 cm, 50.92% increase), and total root length (47.14% increase), with plant height surpassing the suboptimal QFH4 treatment (98.33 cm) by 11.53%. Leaf morphology displayed differential responses: QFH3 increased leaf area and length by over 63% (55.61 cm^2^ and 61.17 cm, respectively), whereas QFH2 yielded the highest leaf width (29.07 mm; 49.54% increase), indicating selective regulatory effects of fertilizer formulations on organ development. These findings demonstrate that optimized microbial fertilizer co-application improves both morphological architecture and biomass allocation patterns in oats, providing critical technical parameters for enhancing forage productivity in alpine regions.

### 3.2. Effects of Microbial Fertilizer Treatments on Photosynthetic Characteristics of Oats

As shown in Table 3, the co-application of microbial fertilizers with livestock manure organic fertilizer significantly enhanced photosynthetic efficiency in *Avena sativa* L. ‘Baiyan No.7’. The relative chlorophyll content (SPAD value) reached a maximum of 61.19 under the BFH3 treatment, representing a 46.28% increase compared with the control (BCK) (*p* < 0.05). The net photosynthetic rate (*Pn*) peaked at 14.86 μmol·m^−2^·s^−1^ under the BEM4 treatment, with a 46.12% increase over BCK (*p* < 0.05). The intercellular CO_2_ concentration (*Ci*) ranged from 375.00 to 443.00 μmol·mol^−1^, with all treatments except BEM1 and BFH1 showing significantly higher *Ci* than BCK (4.29–23.17% increase; *p* < 0.05). Both transpiration rate (*Tr*) and stomatal conductance (*Gs*) achieved maxima under BEM4: 3.88 mmol·m^−2^·s^−1^ and 369.67 mmol·m^−2^·s^−1^, respectively, corresponding to 46.42% and 73.29% increases over BCK (*p* < 0.05).

For *Avena sativa* L. ‘Qingyin No.2’ (Table 4), photosynthetic performance was markedly improved by the microbial fertilizer co-application. The relative chlorophyll content (SPAD value) peaked at 68.04 under the QFH3, representing a 47.75% increase over the control (QCK) (*p* < 0.05), with no significant differences observed among the QFH3, QFH5, QFH4, QEM4, and QFH2 treatments. The net photosynthetic rate (*Pn*) peaked at 14.97 μmol·m^−2^·s^−1^ in the QFH4 treatment, closely followed by the QFH3 treatment, both of which were significantly higher than QCK (*p* < 0.05). The intercellular CO_2_ concentration (*Ci*) reached a maximum of 456.67 μmol·mol^−1^ under the QEM4 treatment, reflecting a 24.32% increase over QCK (*p* < 0.05). The QFH3 treatment exhibited a 40.78% increase in transpiration rate (*Tr*), while the highest stomatal conductance (*Gs*) (383.67 mmol·m^−2^·s^−1^) was observed in the QFH5 treatment, corresponding to a 66.33% improvement (*p* < 0.05).

Collectively, these findings demonstrate that the synergistic effects of microbial fertilizers and organic manure enhance photosynthetic performance by optimizing chloroplast functionality, improving stomatal conductance, and elevating CO_2_ assimilation efficiency.

### 3.3. Effects of Microbial Fertilizer Treatments on Oat Forage Yield

As shown in Figure 3a–c, the co-application of microbial fertilizers with livestock manure organic fertilizer significantly increased fresh and dry forage yields of *Avena sativa* L. ‘Baiyan No.7’ but had no significant effect on the fresh-to-dry ratio. Both fresh and dry forage yields reached their maxima under the BFH3 treatment: 36,797.00 kg·hm^−2^ and 9061.60 kg·hm^−2^, representing 34.53% and 38.64% increases over the control (BCK), respectively (*p* < 0.05). The fresh-to-dry ratio was the lowest in BFH3 (4.06), a 2.87% decrease compared with BCK, with all treatments exhibiting ratios below 4.20.

For *Avena sativa* L. ‘Qingyin No.2’ (Figure 3d–f), microbial fertilizer co-application enhanced both fresh and dry forage yields and increased the fresh-to-dry ratio. The QEM2 treatment yielded the highest fresh and dry forage outputs: 38,073.00 kg·hm^−2^ and 8318.80 kg·hm^−2^, corresponding to 33.43% and 19.77% increases over the control (QCK), respectively (*p* < 0.05). The fresh-to-dry ratio ranged from 4.55 to 4.74 across all treatments, all of which were higher than QCK.

Collectively, these findings demonstrate that the combined application of microbial fertilizers and organic manure significantly improves oat fresh and dry forage yields, with BFH3 and QEM2 identified as the optimal treatments for ‘Baiyan No.7’ and ‘Qingyin No.2’, respectively.

### 3.4. Effects of Microbial Fertilizer Treatments on Nutritional Quality of Oat Forage

As shown in Table 5, the co-application of microbial fertilizers with livestock manure organic fertilizer improved the crude protein (CP), ether extract (EE), and soluble sugar (SS) contents of *Avena sativa* L. ‘Baiyan No.7’, while reducing fiber content and increasing total digestible nutrients (TDN). The crude protein (CP) and ether extract (EE) contents reached maxima under the BEM4 treatment: 11.02% and 4.70%, representing 33.09% and 56.15% increases over the control (BCK), respectively (*p* < 0.05). The highest soluble sugar (SS) content (13.82%) was observed in BEM5, reflecting a 20.09% increase (*p* < 0.05). Both acid detergent fiber (ADF) and neutral detergent fiber (NDF) were minimized in BEM4 (30.71% and 36.12%, respectively), corresponding to 13.01% and 14.67% reductions compared to BCK (*p* < 0.05). The total digestible nutrients (TDN) content peaked at 59.30% in BEM4, a 6.23% increase over BCK (*p* < 0.05).

As shown in Table 6, the co-application of microbial fertilizers with livestock manure organic fertilizer significantly enhanced the nutritional quality of *Avena sativa* L. ‘Qingyin No.2’. Crude protein (CP) and ether extract (EE) contents peaked under the QFH3 treatment: 12.57% and 3.74%, representing 39.20% and 47.24% increases over the control (QCK), respectively (*p* < 0.05). The highest soluble sugar (SS) content (15.58%) was observed in QFH3, a 26.67% increase over QCK. Both acid detergent fiber (ADF) and neutral detergent fiber (NDF) were minimized in QFH3 (30.15% and 39.42%, respectively), corresponding to 9.95% and 64% reductions compared to QCK (*p* < 0.05). The total digestible nutrients (TDN) content reached 59.73% in QFH3, significantly higher than QCK (*p* < 0.05), followed by QFH5 and QEM4. These results demonstrate that microbial fertilizers synergistically improved forage nutritional value and digestibility by reducing fiber content and increasing protein and soluble sugar levels, providing a scientific basis for high-quality oat forage cultivation in alpine regions.

As illustrated in Figure 4a,b, microbial fertilizer co-application significantly enhanced the forage value of ‘Baiyan No.7’. The relative feed value (RFV) was highest in BEM4, BEM5, and BFH3: 167.37, 162.98, and 162.36, respectively, representing 24.12%, 20.86%, and 20.40% increases over BCK (*p* < 0.05), with no significant differences among these treatments. The relative forage quality (RFQ) peaked at 160.20 in BEM4, a 24.53% improvement over BCK (*p* < 0.05), indicating a significant enhancement in overall forage value.

As illustrated in Figure 4c,d, microbial fertilizer co-application significantly enhanced the forage value of ‘Qingyin No.2’. The relative feed value (RFV) peaked at 154.53 in QFH3, a 22.02% increase over QCK (*p* < 0.05). The relative forage quality (RFQ) reached 147.96 in QFH3, a 22.30% improvement over QCK, indicating significant enhancements in nutritional quality and digestibility. Collectively, these findings demonstrate that microbial fertilizers synergistically improved forage quality and animal digestion performance by optimizing fiber content and nutritional composition.

### 3.5. Effects of Microbial Fertilizer Treatments on Soil Nutrients in Oat Forage Fields

As shown in Table 7, the co-application of microbial fertilizers with livestock manure organic fertilizer significantly improved soil fertility in *Avena sativa* L. ‘Baiyan No.7’ forage fields. Organic carbon (SOC) and phosphorus content: the SOC (43.06 g·kg^−1^), total phosphorus (TP, 1.98 g·kg^−1^), and available phosphorus (AP, 47.95 mg·kg^−1^) contents were highest in the BFH3 treatment, representing 28.85%, 32.89%, and 41.62% increases over the control (BCK), respectively (*p* < 0.05). Nitrogen and potassium content: the available nitrogen (AN) and available potassium (AK) contents peaked in the BEM4 treatment, with 52.03% and 14.82% increases over BCK, respectively (*p* < 0.05). The total potassium (TK) content was highest in the BEM5 treatment, significantly exceeding all other treatments (*p* < 0.05). The total nitrogen (TN) content reached its maximum of 4.36 g·kg^−1^ in the BFH2 treatment, a 30.14% increase over BCK (*p* < 0.05).

As shown in Table 8, the co-application of microbial fertilizers with livestock manure organic fertilizer significantly enhanced soil nutrient levels in *Avena sativa* L. ‘Qingyin No.2’ forage fields. Organic carbon (SOC) content peaked at 41.51 g·kg^−1^ in the QEM4 treatment, representing a 26.21% increase over the control (QCK) (*p* < 0.05). Nitrogen and potassium content: the total nitrogen (TN, 4.21 g·kg^−1^), total potassium (TK, 29.48 g·kg^−1^), and available potassium (AK, 473.06 mg·kg^−1^) contents were highest in the QEM3 treatment, corresponding to 29.14%, 28.06%, and 18.82% increases over QCK, respectively (*p* < 0.05). Phosphorus content: the total phosphorus (TP) and available phosphorus (AP) contents reached their maxima in the QFH4 treatment, with 22.58% and 30.09% increases over QCK, respectively (*p* < 0.05). The available nitrogen (AN) content peaked at 274.73 mg·kg^−1^ in the QFH5 treatment, significantly higher than QCK and most other treatments (*p* < 0.05). These findings demonstrate that microbial fertilizers sustainably improved soil fertility by promoting organic matter mineralization and nutrient activation.

### 3.6. Effects of Microbial Fertilizer Treatments on Soil Enzyme Activity and pH in Oat Forage Fields

As shown in Figure 5a,b, the co-application of microbial fertilizers with livestock manure organic fertilizer significantly reduced soil pH and enhanced enzyme activity in *Avena sativa* L. ‘Baiyan No.7’ fields. Soil pH analysis revealed that across the ten microbial fertilizer treatments, soil pH ranged from 8.12 to 8.53, all lower than the control (BCK, 8.63), with reductions of 1.16~5.91%, indicating effective neutralization of soil alkalinity by microbial fertilizers. The activity of β-glucosidase (S-βGC) peaked at 149.03 U·g^−1^ in the BEM4 treatment, representing a 63.86% increase over BCK (*p* < 0.05). N-acetyl-β-D-glucosaminidase (S-NAG) activity reached its maximum of 109.85 U·g^−1^ in BEM3, a 56.30% increase over BCK (*p* < 0.05). Both dehydrogenase (S-DHA) and urease (S-UE) activities were highest in BEM4: 21.63 U·g^−1^ and 2386.22 U·g^−1^, respectively, corresponding to 61.42% and 54.26% increases over BCK (*p* < 0.05). Neutral protease (S-NPT) activity peaked at 0.82 μmol·d^−1^·g^−1^ in BFH3, an 18.84% increase over CK (*p* < 0.05).

As shown in Figure 5c,d, soil pH in *Avena sativa* L. ‘Qingyin No.2’ was lowest in the QEM3 treatment (8.20, a reduction compared to the control (QCK, 8.56) (*p* < 0.05)). The activity of β-glucosidase (S-βGC) peaked at 156.39 U·g^−1^ in QFH5, a 65.16% increase over QCK (*p* < 0.05). N-acetyl-β-D-glucosaminidase (S-NAG) activity reached its maximum of 117.25 U·g^−1^ in QFH3, a 60.35% increase over QCK. Both dehydrogenase (S-DHA) and neutral protease (S-NPT) activities were highest in QFH3, corresponding to 53.33% and 33.82% increases over QCK, respectively (*p* < 0.05). Urease (S-UE) activity peaked at 2187.03 U·g^−1^ in QFH2, a 64.05% increase over QCK (*p* < 0.05). These results demonstrate that microbial fertilizers enhanced soil nutrient transformation capacity by regulating the soil microenvironment and promoting enzyme activity.

### 3.7. Correlation Analysis of Oat Parameters Under Different Microbial Fertilizer Treatments

As illustrated in Figure 6a, 25 pairs of significant (*p* < 0.05) or highly significant (*p* < 0.01) positive correlations were detected among oat agronomic traits and forage yield. Specifically, plant height exhibited highly significant synergistic effects with root length, stem base diameter, and leaf area. Root length showed highly significant positive correlations with stem base leaf area, while stem base diameter was highly positively correlated with leaf length and leaf area. Although the fresh-to-dry ratio displayed a negative trend, it did not reach statistical significance, indicating that dry matter accumulation efficiency is governed by other regulatory pathways.

As shown in Figure 6b, 37 significant correlations were identified among 11 parameters, including 27 positive and 10 negative relationships. Neutral detergent fiber (NDF) and acid detergent fiber (ADF) were significantly positively correlated, (*p* < 0.05) while both exhibited highly significant negative correlations with net photosynthetic rate (*Pn* and total digestible nutrients (TDN) (*p* < 0.01). Photosynthetic- and quality-related indicators demonstrated synergistic effects: relative chlorophyll content (SPAD value) were highly significantly positively correlated with photosynthetic rate (*Pn*) and stomatal conductance (*Gs*) (*p* < 0.01), while photosynthetic rate (*Pn*) formed a cascading enhancement network with intercellular CO_2_ concentration (*Ci*), transpiration rate (*Tr*), stomatal conductance (*Gs*), and crude protein (CP) (*p* < 0.01).

As depicted in Figure 6c, 45 significant correlations were detected among 13 soil parameters, including 39 positive and 7 negative relationships. Soil pH showed highly significant negative correlations with soil organic carbon (SOC), total nitrogen (TN), and total phosphorus (TP) (*p* < 0.01), revealing the inhibitory effects of acidification on soil carbon, nitrogen, and phosphorus pools. The enzyme activity network was centered on β-glucosidase (S-βGC), which exhibited highly significant synergistic effects with N-acetyl-β-D-glucosaminidase (S-NAG), soil dehydrogenase (S-DHA), and neutral protease (S-NPT), driving organic matter transformation (*p* < 0.01).

### 3.8. Principal Component Analysis and Comprehensive Evaluation of Membership Function for Oat Parameters Under Different Microbial Fertilizer Treatments

Principal component analysis (PCA) was performed on 33 indicators related to oat growth, yield, quality, and soil quality, extracting two principal components (Figure 7). In the EM microbial fertilizer treatment group (Figure 7a), the maximum eigenvalue reached 22.47, with the variance contribution rates of principal component 1 (PC1) and principal component 2 (PC2) being 68.1% and 6.4%, respectively, and a cumulative explanatory rate of 74.5%. Comprehensive evaluation using the membership function (Figure 8a) ranked the treatments as follows: BEM4 > BEM5 > BEM3 > BFH3 > BFH4 > BFH5 > BEM2 > BFH2 > BEM1 > BFH1 > BCK. These results indicate that the co-application of microbial fertilizers with livestock manure organic fertilizer promoted the growth and development of *Avena sativa* L. ‘Baiyan No.7’, improved its yield and quality, and enhanced soil nutrient content in forage fields. The BEM4 treatment exhibited the best performance, followed by BEM5, BEM3, and BFH3, with minimal differences among these treatments. The BCK treatment performed poorly.

In the FH microbial fertilizer treatment group (Figure 7b), the two principal components (PC1 and PC2) extracted by PCA had a cumulative explanatory rate of 73.0%, with a maximum eigenvalue of 20.40. The variance contribution rates of PC1 and PC2 were 61.8% and 11.2%, respectively. Comprehensive evaluation using the membership function (Figure 8b) ranked the treatments as follows: QFH3 > QFH4 > QFH5 > QEM4 > QFH2 > QEM5 > QEM3 > QEM2 > QFH1 > QEM1 > QCK. These results demonstrate that the co-application of microbial fertilizers with livestock manure organic fertilizer significantly promoted the growth and development of *Avena sativa* L. ‘Qingyin No.2’, enhanced its yield and quality, and improved soil nutrient content in forage fields. The QFH3 treatment exhibited the best performance, followed by QFH4, QFH5, and QEM4, while the QCK treatment performed the worst.

The principal component analysis (PCA) and membership function evaluation results collectively indicate that the co-application of microbial fertilizers and organic manure has significant synergistic effects on oat growth, yield, quality, and soil quality. BEM4 and QFH3 were identified as the optimal treatments for ‘Baiyan No.7’ and ‘Qingyin No.2’, respectively, providing a scientific basis for the green cultivation of oat forage in alpine regions.

## 4. Discussion

### 4.1. Synergistic Enhancement of Oat Growth and Development by Co-Application of Microbial Fertilizers and Livestock Manure Organic Fertilizer

Studies have demonstrated that the co-application of effective microorganism fertilizer, compound microbial fertilizers, and livestock manure organic fertilizer significantly promoted oat growth and development, particularly in key morphological indices such as plant height, root length, and leaf area. Increasing microbial fertilizer application rates correlated with progressive improvements in these parameters. For instance, the plant height of *Avena sativa* L. ‘Baiyan No.7’ increased by 41.29% under the BEM3 treatment compared to the BCK treatment (*p* < 0.05), while *Avena sativa* L. ‘Qingyin No.2’ exhibited a 50.92% increase in plant height under the QFH3 treatment compared to the QCK treatment (*p* < 0.05). This phenomenon may be attributed to the accelerated decomposition of organic fertilizers (e.g., livestock manure) by microbial inoculants, which sustains nutrient release in forage fields and provides sufficient nutrients for oat growth [17]. These findings align with Nguyen et al. [18], who reported that microbial fertilizers enhance plant growth rates in a dose-dependent manner. Furthermore, the combined application of microbial fertilizers and organic manure enhances photosynthetic efficiency and nutrient transport capacity in plants [19].

As the primary photosynthetic organ in oats, the flag leaf directly influences yield and quality through its structural and functional characteristics. The co-application of effective microorganism fertilizer or compound microbial fertilizers with livestock manure organic fertilizer effectively improved leaf architecture and photosynthetic efficiency, thereby driving dry matter accumulation. This result is consistent with the findings of Marika et al. [20], who demonstrated that microbial fertilizers enhance soil microbial activity, promoting nitrogen, phosphorus, and potassium uptake and subsequently improving leaf growth and photosynthesis. Notably, the EM4 and FH3 treatments significantly enhanced photosynthetic characteristics, whereas EM5 and FH5 inhibited the net photosynthetic rate (*Pn*), suggesting potential “saturation thresholds” or “threshold effects” related to microbial fertilizer dosage or inoculant density. This observation parallels findings in maize studies, where the excessive microbial inoculant application occasionally triggered internal microbial community competition or reduced soil nutrient conversion efficiency [21].

### 4.2. Synergistic Enhancement of Oat Yield and Quality by Co-Application of Microbial Fertilizers and Livestock Manure Organic Fertilizer

The addition of livestock manure organic fertilizer supplements nitrogen, phosphorus, potassium, and other essential elements in forage fields, enriching soil fertility. Further co-application of microbial fertilizers accelerates the decomposition of organic matter, optimizes soil microbial community structure, and enhances oat yield and quality [22]. Studies revealed that the BFH3 treatment yielded the highest fresh and dry forage production for *Avena sativa* L. ‘Baiyan No.7’, with 34.53% and 38.64% increases over BCK, respectively (*p* < 0.05). Similarly, the QEM2 treatment achieved the maximum fresh and dry forage yields for *Avena sativa* L. ‘Qingyin No.2’, showing 29.13% and 25.37% increases over QCK, respectively (*p* < 0.05). These results align with the findings of Ruth et al. [23], who demonstrated that the co-application of microbial fertilizers and organic manure improves soil microbial activity, thereby enhancing nutrient conversion and uptake, ultimately boosting plant yield. Notably, while all treatments exhibited yield improvements, the synergistic effects varied depending on microbial fertilizer dosage and formulation. For instance, the FH3 and EM2 treatments significantly promoted oat yield, whereas FH5 and EM5 slightly inhibited it, further indicating potential “saturation thresholds” or “threshold effects” related to microbial inoculant density or application rates [24].

In terms of nutritional quality, the microbial fertilizer co-application significantly increased crude protein (CP) and ether extract (EE) contents while reducing fiber content. For ‘Baiyan No.7’, the CP and EE contents under the BEM4 treatment reached 11.02% and 4.70%, respectively, compared to 8.28% and 3.01% in BCK. For ‘Qingyin No.2’, the CP and EE contents peaked at 12.57% and 3.74% under the QFH3 treatment, representing 39.20% and 47.24% increases over QCK (*p* < 0.05). Simultaneously, fiber content decreased by 9.95~14.67% across all microbial fertilizer treatments. These findings are consistent with studies on maize forage quality [21], where microbial–organic fertilizer interactions enhanced nitrogen, phosphorus, potassium, and micronutrient uptake, accelerating the synthesis of proteins and soluble sugars while reducing cell wall fiber components, thereby improving overall forage quality.

### 4.3. Effects of Microbial Fertilizer and Livestock Manure Co-Application on Soil Environment in Oat Forage Fields

The soil environment is a critical factor influencing forage yield and quality. This study demonstrated that the co-application of microbial fertilizers and livestock manure organic fertilizer effectively enhanced soil nutrient content while regulating pH and enzyme activity. Consistent with previous studies [17], microbial fertilizers promote the decomposition and transformation of soil organic matter, forming more stable soil aggregates, improving soil aeration and water-holding capacity, and thereby optimizing soil nutrient availability and enzymatic activity. These improvements create favorable physicochemical conditions for root growth. Qi et al. [25] reported that microbial fertilizer application enhances rhizosphere soil properties, increases microbial diversity and abundance, and accelerates organic matter decomposition.

Notably, excessive or imbalanced application of microbial fertilizer may lead to soil nutrient imbalances or pH fluctuations. In this study, the EM1, EM5, FH1, and FH5 treatments exhibited relatively higher soil pH, suggesting that the organic acids produced during microbial metabolism may lower the pH, but the soil buffering capacity is limited. Mismatches between microbial inoculant dosage and nutrient load could induce ecological stress. Bibek [26] and Li et al. [27] observed that microbial fertilizers initially reduced pH but caused a rebound when applied at excessive concentrations, consistent with the trend in this study where soil pH first decreased and then increased with higher microbial fertilizer doses. In high-altitude regions, it is essential to identify region-specific optimal ratios of microbial fertilizers and organic manure to ensure positive contributions to soil carbon–nitrogen cycling while avoiding significant disturbances to soil equilibrium.

### 4.4. Optimization Strategies for Co-Application of Microbial Fertilizers and Livestock Manure Organic Fertilizer

Integrated results from principal component analysis (PCA) and membership function evaluation revealed that the type and dosage of microbial fertilizers differentially influenced oat growth, forage quality, and soil quality. The BEM4 and QFH3 treatments exhibited superior comprehensive performance, whereas EM5 and FH5, despite excelling in specific indices, showed limitations in photosynthetic efficiency or soil nutrient balance. Notably, higher inputs did not necessarily translate into proportional yield improvements. Similar patterns have been observed in high-yield crops like maize, where excessive microbial fertilizer application may lead to soil microbial community imbalances or resource wastage [28,29]. Therefore, region-specific protocols for the co-application of microbial fertilizers and organic manure should be developed based on high-altitude ecological conditions, soil quality, crop requirements, and environmental sustainability objectives. Regular monitoring of soil health and crop growth indicators is critical for dynamically adjusting application strategies to maximize the synergistic effects of microbial-organic fertilizer combinations.

## 5. Conclusions and Prospect

### 5.1. Conclusions

This study demonstrates that the co-application of microbial fertilizers and livestock manure organic fertilizer significantly promotes oat growth and development while enhancing soil nutrient content in alpine ecological regions. The BEM4 treatment (18.75 kg·ha^−1^ of effective microorganism fertilizer + 18,000 kg·ha^−1^ of livestock manure organic fertilizer) yielded the most substantial dry forage production increase for *Avena sativa* L. ‘Baiyan No.7’, reaching 8608.07 kg·ha^−1^, a 31.70% improvement over the control. Similarly, the QFH3 treatment (22.50 kg·ha^−1^ of compound microbial fertilizer + 18,000 kg·ha^−1^ of livestock manure organic fertilizer) achieved optimal dry forage yield for *Avena sativa* L. ‘Qingyin No.2’ at 8255.27 kg·ha^−1^, an 18.86% increase over the control. Furthermore, BEM4 and QFH3 not only boosted oat yield but also enhanced forage nutritional value and improved soil conditions in the forage fields.

In summary, the following optimized fertilization protocols are recommended for oat cultivation in high-altitude regions:For *Avena sativa* L. ‘Baiyan No.7’: apply 18.75 kg·ha^−1^ of effective microorganism fertilizer in combination with 18,000 kg·ha^−1^ of livestock manure organic fertilizer;For *Avena sativa* L. ‘Qingyin No.2’: apply 22.50 kg·ha^−1^ of compound microbial fertilizer in combination with 18,000 kg·ha^−1^ of livestock manure organic fertilizer.

These protocols significantly enhance oat yield, quality, and soil health, providing a scientific foundation and technical framework for efficient oat forage cultivation in high-altitude regions.

### 5.2. Prospect

This study will focus on the following key areas for future development: 1. Validation of result generalizability: repeated experiments will be conducted in other high-altitude regions of the Qinghai-Tibet Plateau (e.g., Naqu, Yushu) to further verify the universality of the findings. These studies will be integrated with climate change models to evaluate the long-term effects and adaptability of the fertilization strategies under varying environmental conditions. 2. Mechanistic insights into microbial–organic fertilizer interactions: future research will delve deeper into the coupling mechanisms between microbial fertilizers and organic manure, with a focus on their sustained impacts on the alpine soil micro-ecosystem. Specific attention will be given to microbial community succession and its interactions with soil carbon and nitrogen cycling processes. 3. Development of a precision nutrient management model for alpine oat cultivation: a precision nutrient regulation model for the oat cultivation system will be established by integrating remote sensing monitoring and machine learning algorithms. This approach aims to optimize the spatial–temporal configuration of fertilization strategies, thereby enhancing nutrient use efficiency and providing a technical framework for the sustainable development of grassland husbandry on the-Tibet Plateau.

## Figures and Tables

**Figure 1 plants-14-01271-f001:**
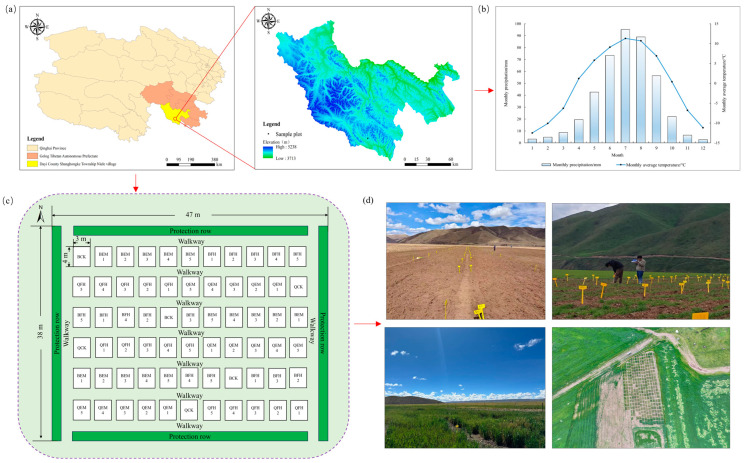
Geographic location of the test site and the layout of the experimental plots in real-life. (**a**) Geographical location map of the experimental site; (**b**) annual climate chart of the experimental site for the year 2024; (**c**) layout diagram of the experimental plots; (**d**) field photograph of the experimental plots.

**Figure 2 plants-14-01271-f002:**
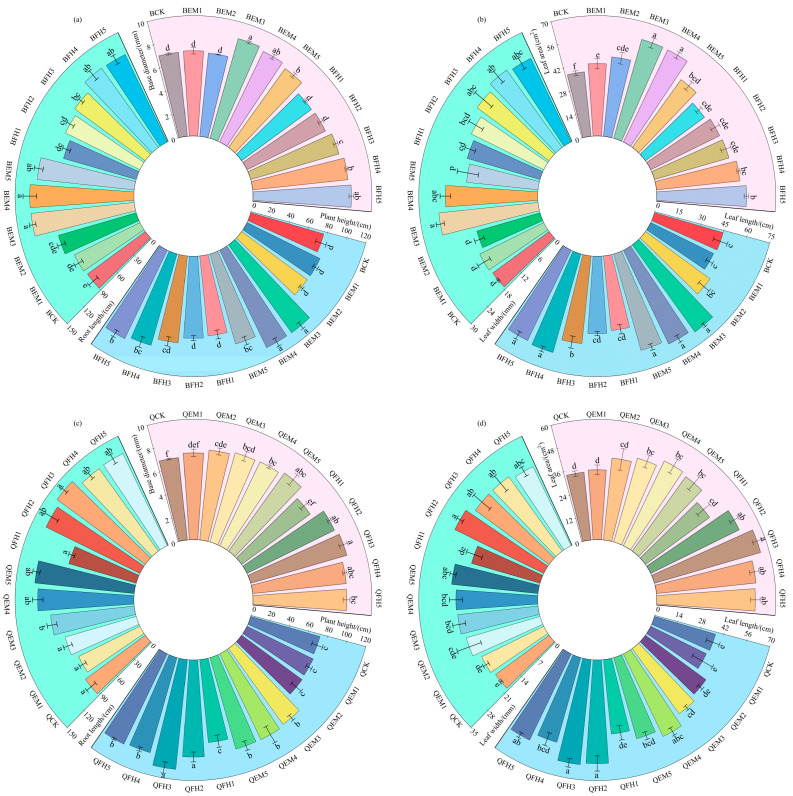
Analysis of oat morphological parameters under different microbial fertilizer treatments. (**a**) Morphological parameters of *Avena sativa* L. ‘Baiyan No.7’: plant height, basal diameter, and root length; (**b**) Leaf traits of *Avena sativa* L. ‘Baiyan No.7’: leaf length, leaf width, and leaf area; (**c**) Morphological parameters of *Avena sativa* L. ‘Qingyin No.2’: plant height, basal diameter, and root length; (**d**) Leaf traits of *Avena sativa* L. ‘Qingyin No.2’: leaf length, leaf width, and leaf area. B: *Avena sativa* L. ‘Baiyan No.7’; Q: *Avena sativa* L. ‘Qingyin No.2’; EM: effective microbial fertilizer (EM); FH: compound microbial fertilizer (FH). Numerical suffixes 1–5 represent gradient levels of microbial fertilizer application. BCK and QCK indicate control treatments (18,000 kg·hm^−2^ livestock manure organic fertilizer without microbial amendments) for *Avena sativa* L. ‘Baiyan No.7’ and ‘Qingyin No.2’, respectively. The experiment was conducted with three biological replicates. Data are presented as mean ± standard deviation, with error bars indicating the standard deviation of the means. Significant differences among fertilizer treatments within the same oat cultivar (*Avena sativa* L.) were assessed using Duncan’s multiple range test at the 0.05 probability level. Lowercase letters (e.g., “a”, “b”) denote statistically distinct groupings across treatments.

**Figure 3 plants-14-01271-f003:**
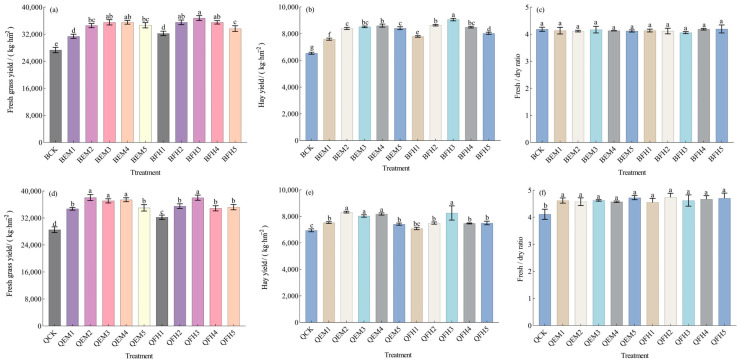
Analysis of oat yield parameters under different microbial fertilizer treatments. (**a**) Fresh grass yield of *Avena sativa* L. ‘Baiyan No.7’; (**b**) Hay yield of *Avena sativa* L. ‘Baiyan No.7’; (**c**) Fresh/dry ratio of *Avena sativa* L. ‘Baiyan No.7’; (**d**) Fresh grass yield of *Avena sativa* L. ‘Qingyin No.2’; (**e**) Hay yield of *Avena sativa* L. ‘Qingyin No.2’; (**f**) Fresh/dry ratio of *Avena sativa* L. ‘Qingyin No.2’. B: *Avena sativa* L. ‘Baiyan No.7’; Q: *Avena sativa* L. ‘Qingyin No.2’; EM: effective microbial fertilizer (EM); FH: compound microbial fertilizer (FH). Numerical suffixes 1–5 represent gradient levels of microbial fertilizer application. BCK and QCK indicate control treatments (18,000 kg·hm^−2^ livestock manure organic fertilizer without microbial amendments) for *Avena sativa* L. ‘Baiyan No.7’ and ‘Qingyin No.2’, respectively. The experiment was conducted with three biological replicates. Data are presented as mean ± standard deviation, with error bars indicating the standard deviation of the means. Significant differences among fertilizer treatments within the same oat cultivar (*Avena sativa* L.) were assessed using Duncan’s multiple range test at the 0.05 probability level. Lowercase letters (e.g., “a” and “b”) denote statistically distinct groupings across treatments.

**Figure 4 plants-14-01271-f004:**
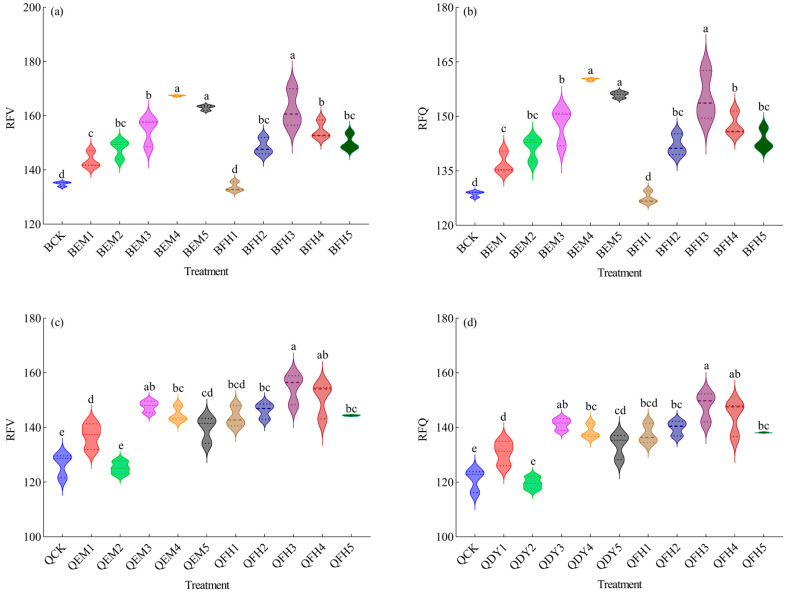
Analysis of relative feeding value (RFV) and relative forage quality (RFQ) of oats under different microbial fertilizer treatments. (**a**) relative feeding value (RFV) of *Avena sativa* L. ‘Baiyan No.7’; (**b**) relative forage quality (RFQ) of *Avena sativa* L. ‘Baiyan No.7’; (**c**) relative feeding value (RFV) of *Avena sativa* L. ‘Baiyan No.7’; (**d**) relative forage quality (RFQ) of *Avena sativa* L. ‘Baiyan No.7’. B: *Avena sativa* L. ‘Baiyan No.7’; Q: *Avena sativa* L. ‘Qingyin No.2’; EM: effective microbial fertilizer (EM); FH: compound microbial fertilizer (FH). Numerical suffixes 1–5 represent gradient levels of microbial fertilizer application. BCK and QCK indicate control treatments (18,000 kg·hm^−2^ livestock manure organic fertilizer without microbial amendments) for *Avena sativa* L. ‘Baiyan No.7’ and ‘Qingyin No.2’, respectively. The experiment was conducted with three biological replicates. Data are presented as mean ± standard deviation, with error bars indicating the standard deviation of the means. Significant differences among fertilizer treatments within the same oat cultivar (*Avena sativa* L.) were assessed using Duncan’s multiple range test at the 0.05 probability level. Lowercase letters (e.g., “a” and “b”) denote statistically distinct groupings across treatments.

**Figure 5 plants-14-01271-f005:**
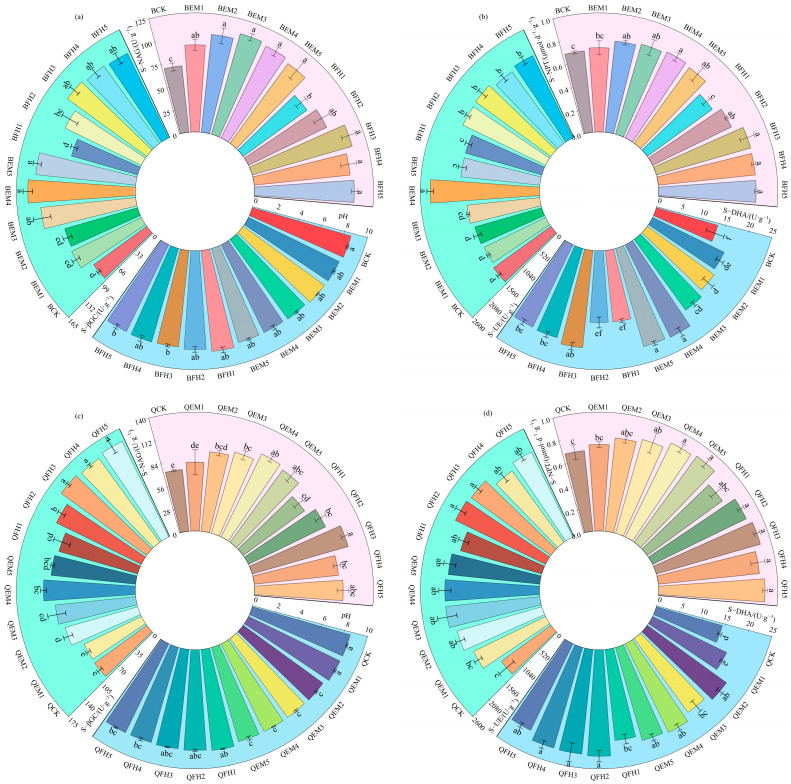
Analysis of rhizosphere soil enzyme activity and soil pH in oats under different microbial fertilizer treatments. (**a**) Rhizosphere soil parameters of *Avena sativa* L. ‘Baiyan No.7’: pH, β-glucosidase (S-βGC), and N-acetyl-β-D-glucosaminidase (S-NAG); (**b**) Rhizosphere soil enzyme activities of *Avena sativa* L. ‘Baiyan No.7’: dehydrogenase (S-DHA), urease (S-UE), and neutral protease (S-NPT); (**c**) Rhizosphere soil parameters of *Avena sativa* L. ‘Qingyin No.2’: pH, β-glucosidase (S-βGC), and N-acetyl-β-D-glucosaminidase (S-NAG); (**d**) Rhizosphere soil enzyme activities of *Avena sativa* L. ‘Qingyin No.2’: dehydrogenase (S-DHA), urease (S-UE), and neutral protease (S-NPT). B: *Avena sativa* L. ‘Baiyan No.7’; Q: *Avena sativa* L. ‘Qingyin No.2’; EM: effective microbial fertilizer (EM); FH: compound microbial fertilizer (FH). Numerical suffixes 1–5 represent gradient levels of microbial fertilizer application. BCK and QCK indicate control treatments (18,000 kg·hm^−2^ livestock manure organic fertilizer without microbial amendments) for *Avena sativa* L. ‘Baiyan No.7’ and ‘Qingyin No.2’, respectively. The experiment was conducted with three biological replicates. Data are presented as mean ± standard deviation, with error bars indicating the standard deviation of the means. Significant differences among fertilizer treatments within the same oat cultivar (*Avena sativa* L.) were assessed using Duncan’s multiple range test at the 0.05 probability level. Lowercase letters (e.g., “a” and “b”) denote statistically distinct groupings across treatments.

**Figure 6 plants-14-01271-f006:**
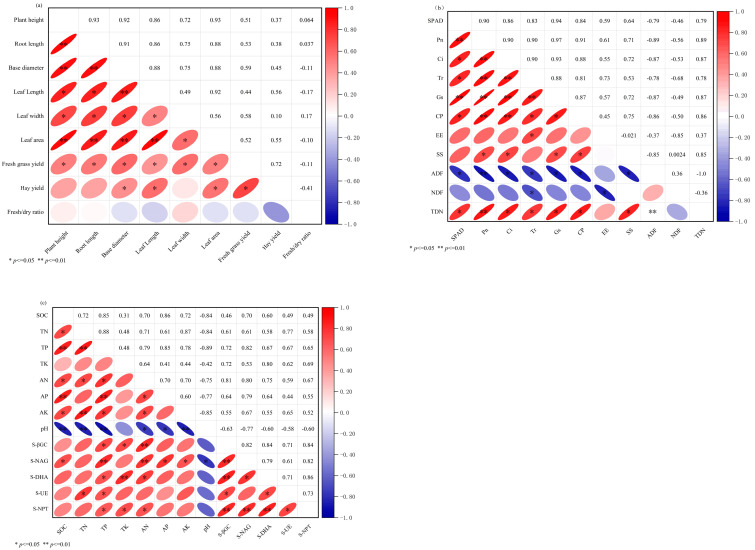
Correlation analysis of oat indexes under different microbial fertilizer treatments. (**a**) Correlation analysis between oat forage yield and agronomic traits; (**b**) correlation analysis between physiological characteristics and nutritional quality of oat forage; (**c**) correlation analysis between soil properties and enzymatic activities in oat rhizosphere. SPAD: relative chlorophyll content; Pn: net photosynthetic rate; Tr: transpiration rate; Gs: stomatal conductance; Ci: intercellular CO_2_ concentration; CP: crude protein; EE: ether extract (crude fat); SS: soluble sugar; NDF: neutral detergent fiber; ADF: acid detergent fiber; SOC: soil organic carbon; TN: total nitrogen; TP: total phosphorus; TK: total potassium; AN: alkali-hydrolyzable nitrogen; AP: available phosphorus; S-βGC: soil β-glucosidase activity; S-NAG: soil N-acetyl-β-D-glucosaminidase activity; S-DHA: soil dehydrogenase activity; S-UE: soil urease activity; S-NPT: soil neutral protease activity.

**Figure 7 plants-14-01271-f007:**
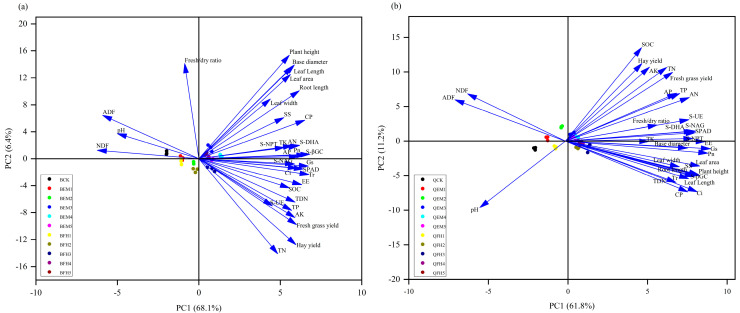
Principal component analysis (PCA) of oat parameters under microbial fertilizer treatments. (**a**) PCA of parameters for *Avena sativa* L. ‘Baiyan No.7’ under different fertilizer applications; (**b**) PCA of parameters for Avena sativa L. ‘Qingyin No.2’ under different fertilizer applications. SPAD: relative chlorophyll content; Pn: net photosynthetic rate; Tr: transpiration rate; Gs: stomatal conductance; Ci: intercellular CO_2_ concentration; CP: crude protein; EE: ether extract (crude fat); SS: soluble sugar; NDF: neutral detergent fiber; ADF: acid detergent fiber; SOC: soil organic carbon; TN: total nitrogen; TP: total phosphorus; TK: total potassium; AN: alkali-hydrolyzable nitrogen; AP: available phosphorus; S-βGC: soil β-glucosidase activity; S-NAG: soil N-acetyl-β-D-glucosaminidase activity; S-DHA: soil dehydrogenase activity; S-UE: soil urease activity; S-NPT: soil neutral protease activity.

**Figure 8 plants-14-01271-f008:**
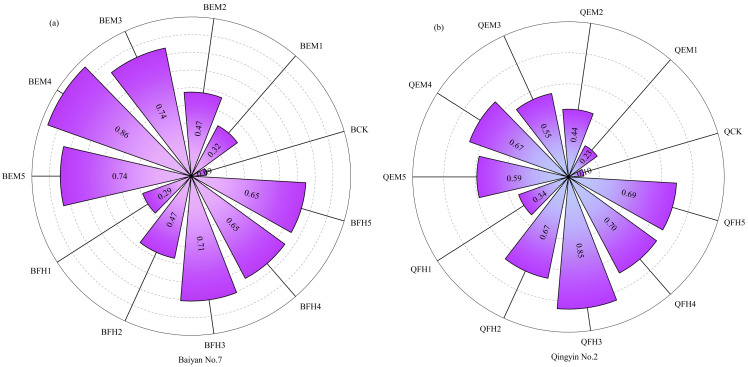
Membership function-based comprehensive evaluation of oat parameters under microbial fertilizer treatments. (**a**) Membership function-based comprehensive evaluation of key indicators for *Avena sativa* L. ‘Baiyan No.7’ across fertilization regimes; (**b**) membership function-based comprehensive evaluation of key indicators for *Avena sativa* L. ‘Qingyin No.2’ across fertilization regimes. B: *Avena sativa* L. ‘Baiyan No.7’; Q: *Avena sativa* L. ‘Qingyin No.2’; EM: effective microbial fertilizer (EM); FH: compound microbial fertilizer (FH). Numerical suffixes 1–5 represent gradient levels of microbial fertilizer application. BCK and QCK indicate control treatments (18,000 kg·hm^−2^ livestock manure organic fertilizer without microbial amendments) for *Avena sativa* L. ‘Baiyan No.7’ and ‘Qingyin No.2’, respectively.

**Table 1 plants-14-01271-t001:** Basic physicochemical properties and soil nutrient contents of the experimental oat forage field.

pH	EC/(μs·cm^−1^)	SOC/(g·kg^−1^)	TN/(g·kg^−1^)	TP/(g·kg^−1^)	TK/(g·kg^−1^)	AN/(mg·kg^−1^)	AP/(mg·kg^−1^)	AK/(mg·kg^−1^)
8.87	705	34.21	3.33	1.52	20.84	177.6	34.11	395.55

**Table 2 plants-14-01271-t002:** Experimental design of integrated microbial inoculant and organic fertilizer co-application treatments. B: *Avena sativa* L. ‘Baiyan No.7’; Q: *Avena sativa* L. ‘Qingyin No.2’. EM: effective microbial fertilizer (EM); FH: compound microbial fertilizer (FH). Numerical suffixes 1–5 indicate gradient levels of microbial fertilizer application. BCK and QCK denote control treatments (18,000 kg·hm^−2^ livestock manure organic fertilizer without microbial amendments) for *Avena sativa* L. ‘Baiyan No.7’ and ‘Qingyin No.2’, respectively.

Treatment	Types of Microbial Agents Microbial Bacterial Fertilizer Types	Application Amount of Microbial Bacterial Fertilizer/(kg·hm^−2^)	Application Amount of Microbial Bacterial Fertilizer in the Experimental Plot/(g)	Application Amount of Cattle and Sheep Manure in the Experimental Plot/(g)
BCK, QCK	__	__	__	21,600
BEM1, QEM1	Effective microbial fertilizer	7.50	9.00	21,600
BEM2, QEM2	Effective microbial fertilizer	11.25	13.50	21,600
BEM3, QEM3	Effective microbial fertilizer	15.00	14.00	21,600
BEM4, QEM4	Effective microbial fertilizer	18.75	22.48	21,600
BEM5, QEM5	Effective microbial fertilizer	22.50	26.98	21,600
BFH1, QFH1	Compound microbial fertilizer	7.50	7.00	21,600
BFH2, QFH2	Compound microbial fertilizer	15.00	18.00	21,600
BFH3, QFH3	Compound microbial fertilizer	22.50	26.98	21,600
BFH4, QFH4	Compound microbial fertilizer	30.00	35.98	21,600
BFH5, QFH5	Compound microbial fertilizer	37.50	45.00	21,600

**Table 3 plants-14-01271-t003:** Photosynthetic traits of *Avena sativa* L. ‘Baiyan No.7’ under different microbial fertilizer treatments. SPAD: relative chlorophyll content; Pn: net photosynthetic rate; Tr: transpiration rate; Gs: stomatal conductance; Ci: intercellular CO_2_ concentration. B: Avena sativa L. ‘Baiyan No.7’; EM: effective microbial fertilizer (EM); FH: compound microbial fertilizer (FH). Numerical suffixes 1–5 represent gradient levels of microbial fertilizer application. BCK and QCK indicate control treatments (18,000 kg·hm^−2^ livestock manure organic fertilizer without microbial amendments) for *Avena sativa* L. ‘Baiyan No.7’ and ‘Qingyin No.2’, respectively. The experiment was conducted with three biological replicates. Data are presented as mean ± standard deviation. Significant differences among fertilizer treatments within the same oat cultivar (*Avena sativa* L.) were assessed using Duncan’s multiple range test at the 0.05 probability level. Lowercase letters (e.g., “a”, “b”) denote statistically distinct groupings across treatments.

Treatment	SPAD	Pn	Ci	Tr	Gs
		/(μmol·m^−2^·s^−1^)	/(μmol·mol^−1^)	/(mmol·m^−2^·s^−1^)	/(mmol·m^−2^·s^−1^)
BCK	41.83 ± 2.85 e	10.17 ± 0.09 f	359.67 ± 8.08 e	2.65 ± 0.06 f	213.33 ± 5.51 d
BEM1	48.33 ± 2.60 d	11.06 ± 0.52 e	380.67 ± 11.93 de	3.01 ± 0.09 e	262.00 ± 14.18 c
BEM2	54.51 ± 1.93 abc	12.77 ± 0.25 d	396 ± 13.08 bcd	3.30 ± 0.07 d	303.67 ± 16.17 b
BEM3	59.19 ± 0.75 ab	14.10 ± 0.38 bc	417.33 ± 13.65 bc	3.69 ± 0.06 abc	333.00 ± 7.21 b
BEM4	60.37 ± 2.87 a	14.46 ± 0.32 ab	424.67 ± 12.34 ab	3.88 ± 0.08 a	369.67 ± 14.05 a
BEM5	57.81 ± 3.56 ab	14.86 ± 0.36 a	403.00 ± 7.00 bcd	3.78 ± 0.09 ab	321.00 ± 12.49 b
BFH1	50.34 ± 1.28 cd	11.04 ± 0.28 e	375.00 ± 8.72 de	3.12 ± 0.08 e	270.67 ± 14.29 c
BFH2	53.18 ± 1.70 bcd	12.84 ± 0.28 d	394.33 ± 14.74 cd	3.53 ± 0.17 c	304.33 ± 10.02 b
BFH3	61.19 ± 3.87 a	13.83 ± 0.2 bc	443.00 ± 10.58 a	3.70 ± 0.09 abc	333.00 ± 10.44 b
BFH4	58.79 ± 2.36 ab	13.72 ± 0.32 c	399.33 ± 15.31 bcd	3.60 ± 0.09 abc	328.00 ± 14.8 b
BFH5	56.36 ± 3.54 abc	13.56 ± 0.40 c	397.00 ± 6.00 bcd	3.66 ± 0.06 abc	320.67 ± 5.51 b

**Table 4 plants-14-01271-t004:** Photosynthetic traits of *Avena sativa* L. ‘Qingyin No.2’ under different microbial fertilizer treatments. SPAD: relative chlorophyll content; Pn: net photosynthetic rate; Tr: transpiration rate; Gs: stomatal conductance; Ci: intercellular CO_2_ concentration. Q: *Avena sativa* L. ‘Qingyin No.2’; EM: effective microbial fertilizer (EM); FH: compound microbial fertilizer (FH). Numerical suffixes 1–5 represent gradient levels of microbial fertilizer application. BCK and QCK indicate control treatments (18,000 kg·hm^−2^ livestock manure organic fertilizer without microbial amendments) for *Avena sativa* L. ‘Baiyan No.7’ and ‘Qingyin No.2’, respectively. The experiment was conducted with three biological replicates. Data are presented as mean ± standard deviation. Significant differences among fertilizer treatments within the same oat cultivar (*Avena sativa* L.) were assessed using Duncan’s multiple range test at the 0.05 probability level. Lowercase letters (e.g., “a” and “b”) denote statistically distinct groupings across treatments.

Treatment	SPAD	Pn	Ci	Tr	Gs
		/(μmol·m^−2^·s^−1^)	/(μmol·mol^−1^)	/(mmol·m^−2^·s^−1^)	/(mmol·m^−2^·s^−1^)
QCK	46.05 ± 1.84 c	10.85 ± 0.15 g	367.33 ± 4.04 e	2.82 ± 0.07 d	230.67 ± 7.37 g
QEM1	49.81 ± 3.97 c	11.54 ± 0.47 f	362.67 ± 16.86 e	2.80 ± 0.12 d	263.00 ± 16.64 f
QEM2	55.91 ± 2.03 b	12.61 ± 0.44 e	379.00 ± 6.25 de	3.04 ± 0.16 cd	299.67 ± 14.74 e
QEM3	58.74 ± 2.84 b	13.53 ± 0.32 cd	403.00 ± 9.17 cd	3.35 ± 0.20 bc	330.67 ± 6.51 d
QEM4	59.70 ± 2.35 b	14.11 ± 0.41 abc	456.67 ± 16.26 a	3.73 ± 0.09 ab	342.00 ± 11.36 cd
QEM5	59.16 ± 3.31 b	13.98 ± 0.29 bc	404.67 ± 8.5 cd	3.05 ± 0.53 cd	346.67 ± 5.13 bcd
QFH1	45.56 ± 3.48 c	12.97 ± 0.39 de	399.67 ± 13.32 cd	3.28 ± 0.09 cd	292.67 ± 15.95 e
QFH2	59.18 ± 4.34 b	14.29 ± 0.61 abc	439.33 ± 11.59 ab	3.78 ± 0.07 ab	349.33 ± 13.58 bcd
QFH3	68.04 ± 1.50 a	14.65 ± 0.41 ab	424.00 ± 15.72 bc	3.97 ± 0.05 a	369.67 ± 9.07 ab
QFH4	61.06 ± 4.62 b	14.97 ± 0.19 a	436.00 ± 7.81 ab	3.87 ± 0.16 a	365.33 ± 8.96 abc
QFH5	61.11 ± 4.18 b	14.57 ± 0.21 ab	432.00 ± 13.45 ab	3.77 ± 0.11 ab	383.67 ± 8.33 a

**Table 5 plants-14-01271-t005:** Nutritional quality of *Avena sativa* L. ‘Baiyan No.7’ under different microbial fertilizer treatments. CP: crude protein; EE: ether extract (crude fat); SS: soluble sugar; NDF: neutral detergent fiber; ADF: acid detergent fiber; TDN: total digestible nutrients. B: *Avena sativa* L. ‘Baiyan No.7’; EM: effective microbial fertilizer (EM); FH: compound microbial fertilizer (FH). Numerical suffixes 1–5 represent gradient levels of microbial fertilizer application. BCK and QCK indicate control treatments (18,000 kg·hm^−2^ livestock manure organic fertilizer without microbial amendments) for *Avena sativa* L. ‘Baiyan No.7’ and ‘Qingyin No.2’, respectively. The experiment was conducted with three biological replicates. Data are presented as mean ± standard deviation. Significant differences among fertilizer treatments within the same oat cultivar (*Avena sativa* L.) were assessed using Duncan’s multiple range test at the 0.05 probability level. Lowercase letters (e.g., “a”, “b”) denote statistically distinct groupings across treatments.

Treatment	CP/(%)	EE/(%)	SS/(%)	ADF/(%)	NDF/(%)	TDN/(%)
BCK	8.28 ± 0.07 d	3.01 ± 0.10 e	10.96 ± 0.10 f	35.34 ± 0.25 a	42.33 ± 0.38 a	55.82 ± 0.19 e
BEM1	8.97 ± 0.20 c	3.26 ± 0.06 e	11.95 ± 0.22 cde	34.52 ± 0.75 ab	40.2 ± 0.68 b	56.44 ± 0.57 de
BEM2	9.47 ± 0.33 b	3.65 ± 0.18 d	12.17 ± 0.19 c	33.52 ± 0.65 abc	39.5 ± 0.61 bc	57.19 ± 0.48 cde
BEM3	10.6 ± 0.28 a	4.00 ± 0.27 cd	12.70 ± 0.27 b	32.98 ± 0.43 bcd	38.05 ± 1.25 cd	57.59 ± 0.33 bcd
BEM4	11.02 ± 0.21 a	4.70 ± 0.19 a	13.63 ± 0.58 a	30.71 ± 0.66 e	36.12 ± 0.23 d	59.30 ± 0.50 a
BEM5	10.87 ± 0.14 a	4.43 ± 0.21 ab	13.82 ± 0.2 a	31.16 ± 1.08 de	36.89 ± 0.58 d	58.96 ± 0.82 ab
BFH1	8.13 ± 0.09 d	3.64 ± 0.10 d	11.49 ± 0.1 de	34.70 ± 1.34 ab	43.05 ± 1.07 a	56.31 ± 1.01 de
BFH2	8.99 ± 0.30 c	3.92 ± 0.15 cd	11.40 ± 0.04 e	32.74 ± 1.11 bcd	39.72 ± 1.12 bc	57.78 ± 0.83 bcd
BFH3	10.73 ± 0.30 a	4.33 ± 0.18 bc	12.06 ± 0.23 cd	32.06 ± 0.94 cde	36.66 ± 1.27 d	58.28 ± 0.71 abc
BFH4	10.65 ± 0.15 a	3.97 ± 0.11 cd	11.99 ± 0.22 cde	33.06 ± 0.8 bcd	38.01 ± 0.48 cd	57.54 ± 0.6 bcd
BFH5	10.47 ± 0.29 a	3.98 ± 0.25 cd	11.91 ± 0.17 cde	33.18 ± 0.7 bcd	39.06 ± 0.51 bc	57.44 ± 0.53 bcd

**Table 6 plants-14-01271-t006:** Analysis of nutritional quality of *Avena sativa* L. ‘Qingyin No.2’ under different microbial fertilizer treatments. CP: crude protein; EE: ether extract (crude fat); SS: Soluble sugar; NDF: neutral detergent fiber; ADF: acid detergent fiber; TDN: total digestible nutrients. Q: *Avena sativa* L. ‘Qingyin No.2’; EM: effective microbial fertilizer (EM); FH: compound microbial fertilizer (FH). Numerical suffixes 1–5 represent gradient levels of microbial fertilizer application. BCK and QCK indicate control treatments (18,000 kg·hm^−2^ livestock manure organic fertilizer without microbial amendments) for *Avena sativa* L. ‘Baiyan No.7’ and ‘Qingyin No.2’, respectively. The experiment was conducted with three biological replicates. Data are presented as mean ± standard deviation. Significant differences among fertilizer treatments within the same oat cultivar (*Avena sativa* L.) were assessed using Duncan’s multiple range test at the 0.05 probability level. Lowercase letters (e.g., “a”, “b”) denote statistically distinct groupings across treatments.

Treatment	CP/(%)	EE/(%)	SS/(%)	ADF/(%)	NDF/(%)	TDN/(%)
QCK	9.03 ± 0.23 f	2.54 ± 0.10 d	12.30 ± 0.16 f	33.48 ± 0.38 ab	46.18 ± 1.61 a	57.21 ± 0.28 bc
QEM1	9.25 ± 0.26 f	2.51 ± 0.04 d	12.75 ± 0.34 f	33.37 ± 0.17 ab	42.78 ± 1.46 b	57.30 ± 0.13 c
QEM2	9.19 ± 0.17 f	2.99 ± 0.22 c	13.44 ± 0.14 e	33.63 ± 0.20 a	46.57 ± 0.86 a	57.1 ± 0.15 bc
QEM3	9.78 ± 0.28 ef	3.41 ± 0.07 ab	13.92 ± 0.24 de	31.78 ± 1.19 abc	40.41 ± 0.25 bc	58.5 ± 0.90 abc
QEM4	11.10 ± 0.75 bc	3.45 ± 0.23 ab	14.52 ± 0.38 bcd	30.78 ± 0.94 c	41.75 ± 1.27 bc	59.25 ± 0.71 a
QEM5	10.94 ± 0.14 bc	3.35 ± 0.08 b	14.18 ± 0.08 bcde	31.58 ± 1.49 bc	42.85 ± 0.70 b	58.64 ± 1.12 ab
QFH1	10.09 ± 0.23 de	2.88 ± 0.12 c	13.98 ± 0.28 cde	32.21 ± 0.98 abc	41.32 ± 1.24 bc	58.18 ± 0.74 abc
QFH2	10.78 ± 0.46 bcd	3.26 ± 0.08 b	14.96 ± 0.40 ab	31.04 ± 1.25 c	41.21 ± 1.07 bc	59.05 ± 0.94 a
QFH3	12.57 ± 0.34 a	3.74 ± 0.25 a	15.58 ± 0.33 a	30.15 ± 1.14 c	39.42 ± 1.71 c	59.73 ± 0.85 a
QFH4	11.55 ± 0.31 b	3.52 ± 0.15 ab	14.90 ± 0.6 abc	31.30 ± 1.73 c	39.89 ± 0.92 bc	58.86 ± 1.30 a
QFH5	10.53 ± 0.42 cd	3.56 ± 0.08 ab	14.54 ± 0.73 bcd	30.70 ± 1.05 c	41.88 ± 0.56 bc	59.31 ± 0.79 a

**Table 7 plants-14-01271-t007:** Analysis of rhizosphere soil nutrients in *Avena sativa* L. ‘Baiyan No.7’ under different microbial fertilizer treatments. SOC: soil organic carbon; TN: total nitrogen; TP: total phosphorus; TK: total potassium; AN: alkali-hydrolyzable nitrogen; AP: available phosphorus. B: *Avena sativa* L. ‘Baiyan No.7’; EM: effective microbial fertilizer (EM); FH: compound microbial fertilizer (FH). Numerical suffixes 1–5 represent gradient levels of microbial fertilizer application. BCK and QCK indicate control treatments (18,000 kg·hm^−2^ livestock manure organic fertilizer without microbial amendments) for *Avena sativa* L. ‘Baiyan No.7’ and ‘Qingyin No.2’, respectively. The experiment was conducted with three biological replicates. Data are presented as mean ± standard deviation. Significant differences among fertilizer treatments within the same oat cultivar (*Avena sativa* L.) were assessed using Duncan’s multiple range test at the 0.05 probability level. Lowercase letters (e.g., “a”and “b”) denote statistically distinct groupings across treatments.

Treatment	SOC/(g·kg^−1^)	TN/(g·kg^−1^)	TP/(g·kg^−1^)	TK/(g·kg^−1^)	AN/(mg·kg^−1^)	AP/(mg·kg^−1^)	AK/(mg·kg^−1^)
BCK	33.42 ± 1.41 c	3.35 ± 0.17 d	1.49 ± 0.07 d	21.51 ± 1.40 c	182.94 ± 9.71 f	33.83 ± 0.76 f	404.67 ± 17.54 e
BEM1	37.82 ± 1.05 b	3.36 ± 0.09 d	1.65 ± 0.08 c	23.10 ± 1.06 bc	230.13 ± 15.45 cd	37.90 ± 0.65 e	426.94 ± 5.95 cd
BEM2	40.26 ± 1.80 ab	3.53 ± 0.09 cd	1.77 ± 0.03 b	23.09 ± 0.89 bc	221.56 ± 4.45 de	45.91 ± 0.65 abc	422.11 ± 10.55 d
BEM3	40.76 ± 1.42 ab	3.78 ± 0.17 bcd	1.89 ± 0.11 ab	24.45 ± 0.51 b	252.59 ± 4.06 b	46.33 ± 0.60 abc	452.01 ± 3.57 ab
BEM4	41.11 ± 2.00 ab	4.18 ± 0.09 ab	1.88 ± 0.07 ab	27.47 ± 0.35 a	278.12 ± 8.81 a	46.21 ± 2.06 abc	464.66 ± 14.14 a
BEM5	41.2 ± 0.42 ab	4.04 ± 0.17 abc	1.83 ± 0.07 abc	27.64 ± 0.71 a	270.78 ± 6.04 a	44.28 ± 0.56 c	462.82 ± 6.73 a
BFH1	39.41 ± 0.92 ab	3.65 ± 0.10 cd	1.73 ± 0.08 bc	20.88 ± 2.01 c	205.44 ± 14.01 e	41.11 ± 1.33 d	431.08 ± 8.52 bcd
BFH2	38.02 ± 0.38 b	4.36 ± 0.46 a	1.88 ± 0.05 ab	22.92 ± 1.91 bc	218.94 ± 5.24 de	39.20 ± 0.65 e	461.7 ± 6.22 a
BFH3	43.06 ± 1.00 a	4.34 ± 0.31 a	1.98 ± 0.05 a	25.65 ± 0.08 ab	246.39 ± 8.95 bc	47.95 ± 0.40 a	456.71 ± 5.72 b
BFH4	42.82 ± 1.06 a	4.01 ± 0.18 abc	1.89 ± 0.11 ab	25.16 ± 0.32 ab	225.01 ± 3.74 de	47.42 ± 0.95 ab	450.72 ± 9.50 ab
BFH5	42.14 ± 2.05 a	3.95 ± 0.10 abc	1.92 ± 0.06 ab	25.02 ± 1.00 ab	238.47 ± 6.48 bcd	45.38 ± 0.17 bc	444.23 ± 4.04 abc

**Table 8 plants-14-01271-t008:** Analysis of rhizosphere soil nutrients in *Avena sativa* L. ‘Qingyin No.2’ under different microbial fertilizer treatments. SOC: soil organic carbon; TN: total nitrogen; TP: total phosphorus; TK: total potassium; AN: alkali-hydrolyzable nitrogen; AP: available phosphorus. Q: *Avena sativa* L. ‘Qingyin No.2’; EM: effective microbial fertilizer (EM); FH: compound microbial fertilizer (FH). Numerical suffixes 1–5 represent gradient levels of microbial fertilizer application. BCK and QCK indicate control treatments (18,000 kg·hm^−2^ livestock manure organic fertilizer without microbial amendments) for *Avena sativa* L. ‘Baiyan No.7’ and ‘Qingyin No.2’, respectively. The experiment was conducted with three biological replicates. Data are presented as mean ± standard deviation. Significant differences among fertilizer treatments within the same oat cultivar (*Avena sativa* L.) were assessed using Duncan’s multiple range test at the 0.05 probability level. Lowercase letters (e.g., “a” and “b”) denote statistically distinct groupings across treatments.

Treatment	SOC/(g·kg^−1^)	TN/(g·kg^−1^)	TP/(g·kg^−1^)	TK/(g·kg^−1^)	AN/(mg·kg^−1^)	AP/(mg·kg^−1^)	AK/(mg·kg^−1^)
QCK	32.89 ± 2.11 d	3.26 ± 0.22 c	1.55 ± 0.08 c	23.02 ± 0.70 e	173.11 ± 4.51 d	34.90 ± 2.23 e	398.12 ± 16.96 d
QEM1	38.21 ± 1.25 bc	3.55 ± 0.06 bc	1.67 ± 0.07 bc	26.01 ± 0.46 bcd	184.58 ± 13.14 d	40.31 ± 0.59 cd	428.94 ± 7.63 c
QEM2	43.96 ± 0.44 a	4.07 ± 0.12 a	1.86 ± 0.01 ab	25.51 ± 0.76 cd	252.91 ± 18.1 abc	43.28 ± 0.73 abc	441.90 ± 8.23 bc
QEM3	40.44 ± 2.73 ab	4.21 ± 0.15 a	1.87 ± 0.05 ab	29.48 ± 1.38 a	257.76 ± 6.04 ab	41.47 ± 0.73 cd	473.06 ± 5.83 a
QEM4	41.51 ± 1.43 ab	4.04 ± 0.14 a	1.83 ± 0.02 ab	24.37 ± 2.30 de	240.75 ± 3.36 bc	42.6 ± 1.11 abc	470.31 ± 5.54 a
QEM5	40.55 ± 1.96 ab	4.12 ± 0.07 a	1.82 ± 0.15 ab	24.68 ± 0.65 de	242.42 ± 8.95 bc	42.17 ± 1.90 bcd	459.96 ± 19.00 ab
QFH1	35.25 ± 1.89 cd	3.54 ± 0.21 bc	1.63 ± 0.04 c	26.50 ± 1.07 bcd	190.08 ± 8.02 d	39.54 ± 0.55 d	430.1 ± 3.76 c
QFH2	38.37 ± 3.22 bc	3.87 ± 0.08 ab	1.82 ± 0.09 ab	27.28 ± 0.43 abc	228.38 ± 14.22 c	39.51 ± 0.66 d	428.64 ± 12.33 c
QFH3	38.23 ± 0.75 bc	3.78 ± 0.16 ab	1.84 ± 0.09 ab	27.92 ± 0.47 abc	251.72 ± 7.51 abc	42.93 ± 1.05 abc	435.21 ± 3.17 bc
QFH4	40.11 ± 0.23 ab	4.07 ± 0.26 a	1.90 ± 0.09 a	28.26 ± 1.24 ab	271.81 ± 3.53 a	45.40 ± 0.33 a	442.47 ± 10.62 bc
QFH5	39.86 ± 1.04 ab	4.03 ± 0.17 a	1.87 ± 0.09 ab	28.39 ± 0.6 ab	274.73 ± 14.08 a	45.00 ± 0.60 ab	441.76 ± 4.56 bc

## Data Availability

The original contributions presented in this study are included in this article; further inquiries can be directed to the corresponding authors.

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
