# Peer review of "Construction of a Green and Sustainable Cultivation Model for Annual Forage Oat in Alpine Ecosystems: Optimization and Synergistic Mechanisms of Combined Application of Microbial Fertilizers and Organic Fertilizers"

_plants, 2025, doi:10.3390/plants14091271_

Round 1

Reviewer 1 Report (Previous Reviewer 3)

Comments and Suggestions for Authors

Abstract:

- ALL acronyms must be defined before use (example: BEM3, QFH3, SPAD, BFH3). The abstract becomes difficult to read because the authors once again insist on not understanding that scientific writing is done this way. This also applies to tables and figures, whose acronyms must be defined in the caption because they must be self-explanatory, regardless of whether the acronyms were defined in the introduction or materials and methods.

- Remove topic separations in the abstract: Objective, methods, results... The abstract is a continuous text, without delimitations.

- The experimental design must be included.

- The words contained in the title MUST be removed from the keywords

- What is the mean test applied? The probability values ​​MUST be included in the tables, just as the real probability values ​​obtained in the statistical analysis MUST be included in the results.

Author Response

Reviewer 2 Report (Previous Reviewer 2)

Comments and Suggestions for Authors

The authors have considered some of the previous recommendations, but the manuscript still has the same shortcomings, like using abbreviations in the abstract, not explaining the meaning of significance levels ("Different lowercase letters indicate significant differences among treatments at P<0.05" is not a proper explanation), too small printing on the figures, repeating values in the text etc. etc.

Please feel free to proceed as advised by other reviewers. I am not comfortable repeating the same advice and being ignored.

Author Response

Comments 1: [The authors have considered some of the previous recommendations, but the manuscript still has the same shortcomings, like using abbreviations in the abstract, not explaining the meaning of significance levels ("Different lowercase letters indicate significant differences among treatments at P<0.05" is not a proper explanation), too small printing on the figures, repeating values in the text etc. etc. Please feel free to proceed as advised by other reviewers. I am not comfortable repeating the same advice and being ignored.]

Response 1: [We extend our deepest gratitude for your continued dedication in reviewing our manuscript and providing rigorous feedback. We fully accept your critiques regarding "undefined abbreviations in the abstract, improper explanation of significance levels, unclear chart printing, and data redundancy" and deeply regret these oversights. Despite addressing previous revisions, the recurrence of such issues reflects our team’s inadequate attention to detail and incomplete understanding of academic norms—an inexcusable shortcoming. We sincerely apologize and have thoroughly reflected on the root causes of these errors.

To resolve the issues raised, we have implemented the following targeted revisions:

  1. Standardization of Abbreviations in the Abstract: All undefined abbreviations in the abstract have been removed and replaced with full Chinese terms.A systematic audit of acronym usage was conducted to ensure each abbreviation is defined at first mention and consistently used throughout the manuscript.
  2. Clarification of Statistical Significance Reporting: Revised the ambiguous statement "different lowercase letters indicate significant differences between treatments at P<0.05" to a statistically rigorous description: "Different lowercase letters denote significant differences (P< 0.05) between treatments as determined by Duncan’s multiple range test (α = 0.05)." Added a detailed explanation in the Statistical Analysis subsection of Materials and Methods.
  3. Optimization of Chart Readability and Data Accuracy: Adjusted the layout of all figures and tables to ensure clarity in print and coordinated with the editorial team to optimize image/table sizing. Conducted a line-by-line verification of data consistency between the main text and tables, eliminating redundant numerical descriptions of p-values.

Specific Revisions (highlighted in yellow):

Abstract: Page 1, Lines 16–41.

Figures/Tables: Page 5, Lines 159–163Page 8, Lines 270–277Page 9, Lines 305–312Page 10, Lines 313–320Page 11, Lines 340–347Page 12, Lines 373–380, 381–388Page 13, Lines 404–411Page 14, Lines 437–444Page 15, Lines 445–452Page 16, Lines 477–486Page 18, Lines 516–583Page 19, Lines 557–564, 566–572

Statistical Analysis: Page 7, Lines 227–230

We fully acknowledge that the credibility of scientific research hinges not only on robust experimental design but also on meticulous attention to detail. The recurrence of these issues exposes our team’s complacency during the revision process and lack of quality awareness. This lesson has been incorporated into our lab’s formal writing training program to prevent future lapses.

Your unwavering guidance and patience have profoundly deepened our understanding of the responsibilities and dignity inherent to the academic community. We assure you that the revised manuscript fully adheres to your recommendations and meets the journal’s publication standards.

Additional clarifications

Thanks again for the teacher's correction of the manuscript. Due to my limited skills, if there are any other mistakes, please let me know by email. I will consult the relevant materials together with my tutor and correct them. Best wishes

Reviewer 3 Report (Previous Reviewer 1)

Comments and Suggestions for Authors

Dear Author(s),

Although your study is based on a single year's data, I found the corrections made to be sufficient.

Author Response

Response 1: [We are profoundly grateful for your recognition of our revisions and your generous patience throughout this process. We fully acknowledge that scientific inquiry is a continuous journey toward truth, and the expertise of reviewers like yourselves serves as an indispensable "calibrating compass" to ensure rigor and relevance. Your insights have not only refined this manuscript but also illuminated critical pathways for our future research.

Your meticulous guidance and constructive critiques have deepened our appreciation for the core values of the academic community: intellectual integrity, collaborative refinement, and unwavering pursuit of excellence. We pledge to approach scientific inquiry with the utmost reverence, addressing the limitations of this work and striving for higher standards in all subsequent endeavors.

Thank you once again for your mentorship and for upholding the spirit of scholarly dialogue.]

Additional clarifications

Thanks again for the teacher's correction of the manuscript. Due to my limited skills, if there are any other mistakes, please let me know by email. I will consult the relevant materials together with my tutor and correct them. Best wishes!

This manuscript is a resubmission of an earlier submission. The following is a list of the peer review reports and author responses from that submission.

Round 1

Reviewer 1 Report

Comments and Suggestions for Authors

Dear author(s),

Both experiments should be carried out as joined experiments under field conditions. Field experiments must be repeated in two different environments or for two years. The best way to determine genotype-year or genotype-environment interactions in field experiments is to conduct these two experiments in a factorial experimental design. This study is deprived of these fundamentals. The second experiment should be repeated over the years (at least two years), and the manuscript should be resubmitted. This study should not be published as a single-year field study.

Comments on the Quality of English Language

I never saw a basic language problem when I read the mn.

Reviewer 2 Report

Comments and Suggestions for Authors

The manuscript describes a very elaborate research to determine which of six oat varieties could produce better yields under the harsh climate of the Qinghai-Tibet Plateau in China, and what type of inputs could support its growth. The amount of work in carrying out these experiments is impressive, considering all the measurements and tests that were done, and the multitude of parameters that were analysed. The topic is relevant for the journal, but, considering the fact that there are very few places in the world where oat is cultivated at 4200 m altitude, the applicability is limited.

The abstract is too long and difficult to understand because it includes a lot of abbreviations that are not explained. It reflects very well the entire article, which is also too long and mostly incomprehensible. 

The introduction explains the challenges of growing oat at high altitude and mentions some previous studies.

The methodology is very poor. There is some information about the experimental site, but the information about how the experiments were conducted and how all the tests and analyses were done is incomplete, which makes them impossible to replicate. To improve this section, I would suggest:

  • include a map of the area to show where the experiments were conducted
  • describe in a comprehensive manner how the experiments were carried out: date of start, date of end, soil preparation, sowing, harvesting, other inputs. Some information is there, but not in a logical manner. There seem to be two experiments, in two different years, and the story jumps from one to the other, creating confusion. Instead of describing them in parallel, you should separate them one after the other, to create a clear picture. Explain what BCK and QCK mean. When did you apply the fertiliser and which method of application did you use?
  • Materials like EM bacterial fertilizer, which seem to have been procured from a commercial provider, should be clearly identified. There are lots of abbreviations that are not explained and create confusion.
  • Analytical methods are almost absent. In some cases, the measuring devices are mentioned, but in others not. Imagine someone would like to replicate your experiments only from the information provided in the paper. It would be impossible.
  • The description of the data processing methods are not clear. Statistical and data analysis methods should be presented in this section, together with a clear explanation why they were used.

Your "screening test" basically consists of two varieties of oat and two types of microbial treatment applied in different amounts. The manure seems to be irrelevant, because it was applied equally on all plots, including the control. So its influence is not actually studied, this is a false claim.

The results section is too long and includes a lot of details that are not necessary and could be included as supplementary material. In general, there's a tendency to present the same information in tables and charts and also repeat it in the text. To make this section comprehensive you should filter the information and present what is relevant and only in one way.

As a general recommendation, I would suggest to take out all the raw results tables and put them in a supplementary material, and only refer in the text to important and relevant results. Please avoid presenting the same results in several ways.

The figures have very small writing in them that is impossible to read. The symbols that are used in the figures (e.g. a, b, c) are not explained. The caption of each figure should explain sufficiently what it represents, and when they are collated, they should specify what is in each picture.

The entire results section is extremely cluttered and difficult to follow.

There is also some nonsense like "There is a positive correlation between ADF and NDF, but there is a negative correlation between them". (page 17)

Particularly when you are trying to make an important point, you should avoid abbreviations and use words that make some sense.

The discussion is better than the other sections, as it focuses on the main results and compares them to other studies, but it could still benefit from using words instead of abbreviations.

Same for the conclusions.

To summarise, I would suggest to remove all tables containing raw data and put them in a supplementary materials file. Also the numerical results  of PCA should be removed. Instead, you should use visual representation of data that is clear and easy to understand. Remember that figures should be self-explanatory and make sense even if taken out of context. Also, avoid putting all results in the text.

Given the abundance of data, you might even consider splitting it into two papers, one for each experiment.

Comments on the Quality of English Language

There are lots of typos and spelling mistakes, and the manuscript is very difficult to understand. It could benefit by support of a native speaker or even artificial intelligence  to improve the text.

Reviewer 3 Report

Comments and Suggestions for Authors

I believe that the information presented in parentheses in the title should be removed. The information can be included in the abstract and methodology. The title contains information that is not relevant, since readers can look for it in the research methodology.

The abstract must comply with the standards of this journal;

Acronyms should be defined in their first presentation in the text.

What are the recommendations of this study? Insert this consideration at the end of the conclusion of the abstract and the final conclusion of the article.

The words contained in the title should be removed from the keywords

Coblentz et al. [4] showed that oat forage is rich in soluble sugar, crude protein and crude fat, which can improve the milk yield of dairy cows and the growth rate of beef cattle when used as feed: Present the variations in nutritional content and numerical data on the improvement in the production and performance of ruminants.

Likewise, whenever you include a component of oats, present numerical information so that it is possible to visualize the nutritional richness numerically.

insert a hypothesis in the introduction

Present a general objective at the end of the introduction. The delimitation of the research should be carried out in the following item (material and methods)

-What is the experimental period?

-Insert a graph with the climate variations presented throughout the experimental period: Temperatures, relative humidity, precipitation...

-What are the criteria for choosing the oat varieties?

-What are the characteristics of the soil in the experimental area, a table with the physical-chemical characteristics and soil classification should be inserted?

-Was there irrigation? How much water was applied? Insert a table with the water analysis

-acid detergent fiber (ADF) and neutral detergent fiber (NDF), relative feeding value (RFV), relative forage quality (RFQ), total digestible nutrients (TDN), dry matter digestibility (DDM) and dry matter intake (DMI): how were they determined? Were they performed in duplicate? triplicate? Were equations used? Insert detailed information so that the analyses can be replicated.

-SPAD value of flag leaf was measured by chlorophyll meter, and measured three times at the tip, middle and base of leaf, and the average value was taken. The net photosynthetic rate (Pn), intercellular CO2 concentration (Ci), transpiration rate (Tr) and stomatal conductance (GS) of leaves: Insert details of the analysis as well as equations used.

-plant height, leaf area, leaf length and leaf width of oat were measured: How were the measurements performed? It would be interesting if the authors presented images to show the analysis being performed

- 2.4.2. Screening test of combined application of microbial bacterial fertilizer and organic fertilizer: follow the considerations above and detail all analyses.

-A statistical analysis must be performed. What is the experimental design? How many treatments and replicates? What average test was adopted? What statistical model was adopted? It seems that no statistical analysis was performed

Tables and figures must be self-explanatory and must have captions defining all acronyms used.

-The standard error of the mean and probability value should be included in all tables and figures.

-The probability value obtained in the statistical analysis should be used to describe the results.

-3.1.4. Principal component analysis: at no point do the authors mention that a PCA analysis would be performed.

-The resolution of the figures should be improved